# Phyllosphere microbial communities are modulated by pathogen coinfection, but not a plant defense hormone

**Julie K. Geyer**[1,2]*, **Rita L. Grunberg**[1,3], **Charles E. Mitchell**[1,4]

1 Department of Biology, The University of North Carolina at Chapel Hill, Chapel Hill, North Carolina, United States of America, 2 Department of Pathology and Laboratory Medicine, The University of North Carolina at Chapel Hill, Chapel Hill, North Carolina, United States of America, 3 Wilson Center for Science and Justice at Duke Law, Durham, North Carolina, United States of America, 4 Environment, Ecology, and Energy Program, The University of North Carolina at Chapel Hill, Chapel Hill, North Carolina, United States of America

* jgeyer@email.unc.edu

## Abstract

Phyllosphere microbial communities play important roles in plant health, yet the roles of plant defense hormones and coinfections in shaping these communities remain unclear. This study investigated how exogenous application of the plant defense hormone salicylic acid and fungal coinfection influenced microbial communities on leaves of tall fescue. In a factorial experiment, we treated leaves with salicylic acid (at 100 mg/L) or a control solution and inoculated them with one of four inoculation treatments: *Rhizoctonia solani* alone, mock inoculation, co-inoculation with both *R. solani* and *Colletotrichum cereale*, or mock co-inoculation. We characterized the fungal and bacterial communities using ITS and 16S rRNA gene sequencing, respectively. Salicylic acid application did not significantly alter the diversity, composition, or taxa abundances of either fungal or bacterial communities. In contrast, co-inoculation significantly shifted fungal community composition and increased fungal diversity compared to inoculation with *R. solani* alone. Bacterial communities were not significantly impacted by either inoculation treatment. These results suggest that in this system, coinfection has a stronger influence on phyllosphere fungal communities than exogenous salicylic acid application. Our findings highlight the potential for pathogen coinfections to shape plant-associated microbial communities, particularly fungi, and emphasize the need for further research on the effects of salicylic acid across different host species and experimental approaches.

## Introduction

Plant defenses are activated by several intrinsically produced hormones, like salicylic acid and jasmonic acid, which protect the host against disease and herbivory [1–3]. These hormones act within a broader multi-layered plant immune system including

**Data availability statement:** Sequencing Datasets: https://www.ncbi.nlm.nih.gov/bioproject/PRJNA1256440.

**Funding:** USDA-NIFA, Grant/Award Number: 2016-67013-25762, https://www.nifa.usda.gov/ (CEM) ; National Science Foundation, Grant/Award Number: DEB-2308472, https://www.nsf.gov/ (CEM). Neither funding agency was involved in study design, data collection and analysis, decision to publish, or preparation of the manuscript.

**Competing interests:** The authors have declared that no competing interests exist.

innate physical defenses, and intracellular proteins [4]. Many studies have shown that salicylic acid influences a plant's susceptibility to certain diseases and can be especially protective against biotrophic pathogens [5]. Furthermore, salicylic acid plays a role in protecting a plant host from future infections through systemic acquired resistance [6,7].

The relationship between salicylic acid, pathogens, and plant-associated microbes is bidirectional and complex. Beyond its role in plant immunity, salicylic acid can both be inhibited by microbes [8] and produced by microbes [9]. For example, endophytic symbionts in grasses can suppress salicylic acid production [10,11], and bacterial [12] and fungal [13] plant pathogens can inactivate salicylic acid. Likewise, mutualistic *Pseudomonas sp.* can produce salicylic acid in plants [14]. Meanwhile, salicylic acid has been shown to influence bacteria on plant roots and leaves [15,16]. Plants can also modify their microbial communities during foliar pathogen attack by releasing exudates into the soil which select for certain beneficial microbes [17–20]– these microbes may indirectly benefit the plant by increasing nutrient uptake or water availability or may directly benefit the plant through competitive inhibition of pathogens [21].

In addition to modulating the interaction between a host and a single pathogen species, salicylic acid has also been shown to modify interactions among co-infecting pathogens. For example, foliar application of salicylic acid can reduce the incidence of coinfections by foliar fungal pathogens by inhibiting biotrophic pathogens [5]. The effect that salicylic acid has on pathogens that are not feeding as biotrophs is thought to relate to antagonism between co-occurring hormone pathways – jasmonic acid is thought to inhibit necrotrophs and salicylic acid production, whereas salicylic acid is thought to inhibit biotrophs and jasmonic acid production [22]. While relatively few studies have examined how plant hormones shape microbial communities during pathogen coinfection, mammalian studies have demonstrated that the host immune system can alter microbial communities differently in pathogen coinfections compared to single infections [23]. In plants, fungal pathogens with different feeding strategies upregulate distinct immune hormone pathways suggesting similar context-dependent effects may occur [4].

Although there are a growing number of studies showing that salicylic acid modifies bacterial communities, to our knowledge, no studies consider the effect of salicylic acid on fungal communities. Likewise, the demonstrable effect of infection on microbial communities has been established, both in plant and human systems, but the effect of coinfection is less well studied. To help fill these gaps, we sought to understand whether coinfection differentially modifies microbial communities compared with single infections, and how that interacts with or compares to the effects of salicylic acid.

To address these questions, we used a factorial experiment on tall fescue plants crossing two treatments: application of salicylic acid or sterile water (control) and pathogen inoculation – co-inoculation with *C. cereale* and *R. solani*, mock co-inoculation (control), single inoculation of *R. solani*, or mock single inoculation (control). Both *R. solani* and *C. cereale* are naturally co-occurring and co-infecting

pathogens of tall fescue. We assessed the causal role of *R. solani* and *C. cereale* fungal pathogens, as well as salicylic acid, on fungal and bacterial diversity, community structure, and differential abundance.

## Materials and methods

To understand how hormone treatment and foliar pathogen infection influenced phyllosphere microbial communities, we ran a factorial manipulation of hormone and inoculation treatment with two hormone treatments and four inoculation treatments. Hormone treatment included two treatment groups: foliar application of salicylic acid and foliar application of sterile water (as a control treatment). Inoculation treatments included four groups: a single inoculation of *R. solani*, a mock single inoculation of *R. solani*, a co-inoculation of *R. solani* and *C. cereale*, and a mock co-inoculation of *R. solani* and *C. cereale*. Inoculation controls allowed us to isolate the effects of each pathogen treatment from procedural (inoculation) effects. We were unable to include a planned single inoculation of *C. cereale* because of a low quantity of *C. cereale* inoculum, and thus also omitted a mock single inoculation of *C. cereale* ([Fig 1]).

### Plant germination and growth

Tall fescue commonly harbors *Epichloë coenophiala*, a systemic, vertically transmitted, fungal endophyte that establishes symbiotic relationships within the above-ground tissues of temperate grasses [24]. To avoid potential variability in *Epichloë*

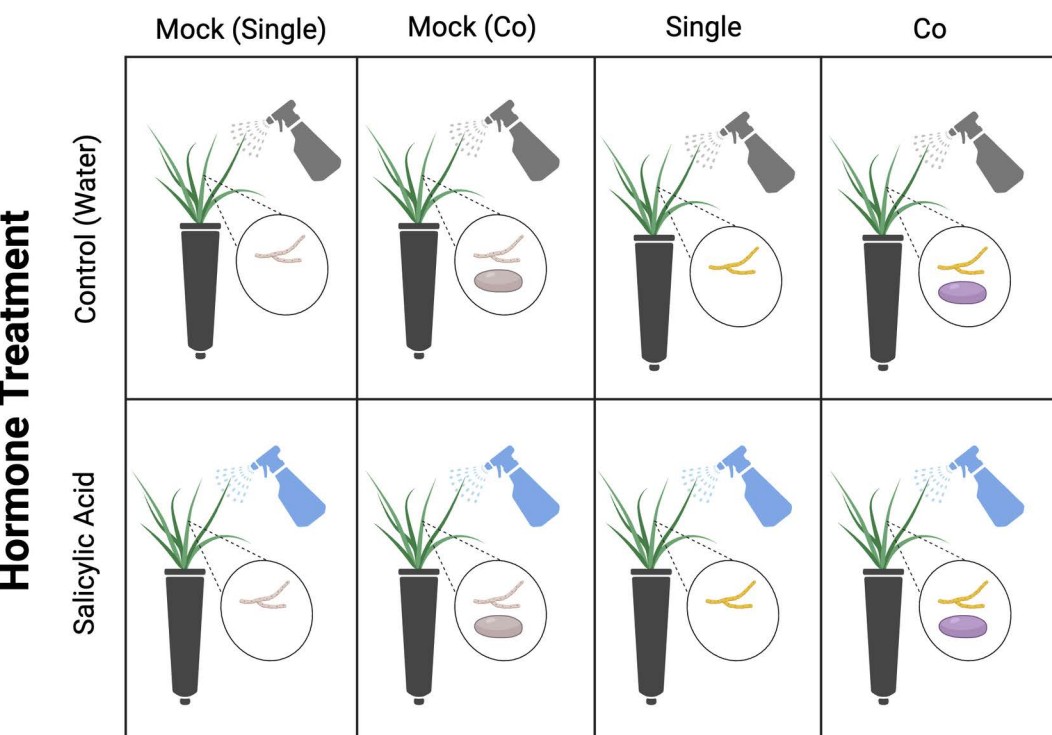

**Fig 1. Methodological overview of the factorial manipulation of hormone treatment and inoculation treatment.** Abbreviations are as follows: "Mock (Single)" = plants that received a mock inoculation of *R. solani*; "Mock (Co)" = plants that received a mock co-inoculation of *R. solani* and *C. cereale*; "Single" = plants that received an inoculation of *R. solani*; "Co" = plants that received a co-inoculation of *R. solani* and *C. cereale*. Plants across inoculation treatment groups either received a hormone treatment of sterile water (control), or salicylic acid.

infection among individual plants, we used tall fescue seeds that were *Epichloë*-free. These seeds were produced from an *Epichloë*-free line of plants derived from a single cultivar, KY-31, by Dr. Tim Phillips at the University of Kentucky, who provided the seeds to us.

To optimize germination rates, seeds were primed by immersion in sterile water for 4 hours, followed by overnight drying in a sterile biosafety cabinet. This priming technique helps to initiate metabolic processes associated with germination, potentially leading to more uniform and rapid seedling emergence. Following priming, we surface-sterilized the seeds to minimize potential microbial contamination [25]. This was achieved by agitating the seeds for 12 minutes in a solution containing 20% bleach and 1% Tween 20, a surfactant that enhances the sterilizing effect. The seeds were rinsed with sterile water to remove any residual sterilizing solution.

For planting, we used autoclaved Sungro Metromix 360 soil, ensuring a sterile growth medium. Soil was sterilized twice, 24 hours apart, for 60 minutes at 121°C. Individual seeds were placed in conical plastic pots (Stuewe and Sons Deepots, Corvallis, OR) measuring 5.08 x 17.78 cm (2 x 7 inches). To provide a consistent, slow-release nutrient source, five Osmocote beads were added to each pot, each containing a single seed.

To support plant growth, we saturated the soil with water three times a week, and the plants were kept under full-spectrum, high-pressure sodium lights from 9:00am to 7:00 pm, supplemented by natural light. The greenhouse conditions were maintained at 70.0°F (21°C), with full-spectrum lights turning on when the natural light level fell below 350 W/m² and turning off when it exceeded 600 W/m². The plants were monitored daily until the second leaf emerged, approximately one month after the seeds were primed and planted, at which point the leaves were marked at the base with a leaf tag. The second leaf was chosen for all inoculation challenges because the second leaf (unlike the first leaf) has the capacity to grow into a mature leaf with complete structural and functional characteristics.

## Hormone treatment

To investigate the influence of salicylic acid on the plant leaf microbiome and its role in modulating interactions between coinfecting pathogens, we treated 6-week-old plants with either an aqueous solution of salicylic acid (at 100 mg/L), or sterile water as a control treatment. Hormone and control treatments were applied to both sides of each leaf until they were visibly saturated. Foliar salicylic application has demonstrated biological activity across multiple host species, including other grass species [3,26], with salicylic acid concentrations as low as 0.1mM [27,28] and as high as 8mM [29]. Hormone treatment intervals and application methods were chosen based on Souri & Tohidloo [30]. Prior to hormone treatments, all plants were relocated from the greenhouse to two growth chambers (Percival PGC- 6L; Perry, Iowa). The temperature in the growth chambers was set to 25°C with a 12-hour dark/12-hour light schedule. 57 plants were treated with a solution of ~ 0.72mM (100 mg/L) salicylic acid, and 59 plants were treated with sterile water as a control treatment. These treatments were repeated two days later following Souri & Tohidloo [30].

## Fungal culturing and inoculations

*R. solani* cultures were revived by placing a single 6mm culture plug (stored in 20% glycerol and potato dextrose broth at −80°C) onto separate petri dishes with sterile potato dextrose agar (hereafter, PDA). Both *R. solani* and *C. cereale* were collected in 2015 from Widener Farm in the Duke Forest Teaching and Research Laboratory. *C. cereale* cultures were revived by placing a single 6mm plug from culture slants (stored at 4°C) onto separate PDA plates. Each petri dish was secured with parafilm, and stored in an incubator set to 28°C. After several weeks of growth, the cultures of *C. cereale* and *R. solani* had reached suitable growth stages for infection challenges – meaning that the leading edge of each culture had nearly reached the perimeter of the PDA plate.

Plants received a factorial manipulation of salicylic acid (detailed above) and inoculation (Fig 1). All 116 tall fescue plants were randomly assigned an infection treatment group – 35 plants received a mock inoculation of *R. solani* (hereafter, mock (single)), 37 plants received an inoculation of *R. solani* (hereafter, single), 22 plants received a

mock inoculation of *R. solani* and *C. cereale* (hereafter, mock (co)), and 22 plants received an inoculation of *R. solani* and *C. cereale* (hereafter, co). The low volume of *C. cereale* spores from lab cultures limited the number of possible co-inoculations. All inoculations (infection challenges) occurred one day after the second (final) salicylic acid application.

Before *C. cereale* inoculation challenges, culture plates were flooded with 10mL of sterile water and gently scraped with a sterile metal loop (to release conidiospores). The released spores were then quantified using a hemocytometer and concentrated to a spore density of 1,000,000 spores/mL. *C. cereale* pathogen challenges were conducted by brushing both sides of the second leaf with 0.5mL inoculum. This method of experimental inoculation was selected to be similar to *C. cereale*'s natural mode of transmission via rain-splashed conidiospores. The second leaf was chosen for inoculation challenges and sampling as it represents the first leaf on a tiller that has the capacity to mature into an adult leaf. Mock *C. cereale* inoculum contained a solution of broth and sterile water. *R. solani* inoculations were conducted by placing a 0.5 cm mycelial plug from the culture plate onto the second leaf with a 2 cm cotton pad soaked in sterile water (to increase humidity and facilitate infection). This method of experimental inoculation was selected to be similar to *R. solani*'s natural mode of transmission via contact of leaves with fungal hyphae. The *R. solani* plug and cotton pad were secured with aluminum foil. Mock *R. solani* plugs contained solidified sterile agar. All co-inoculations and single inoculations occurred on the same leaf of each plant, and the order in which inoculation treatments occurred was randomized. Dew chambers were placed over the plant after inoculation for two days to maintain the humidity necessary for infection, and plants were randomized both within and across the two growth chambers used for the experiment. During the time that dew chambers covered plants, a humidifier was placed on each of the two shelves of each growth chamber to maintain relative humidity at 85−90%, and the temperature in the growth chambers was raised to 28°C – conditions which facilitate establishment of inoculated pathogens. After dew chambers, culture plugs, aluminum foil, and cotton pads were removed, humidifiers were switched off, and the temperature in the growth chamber was returned to 25°C. Disease progression for all plants was assessed daily. Symptomatic infection for each parasite is visually distinct: *R. solani* is a necrotroph which produces light brown lesions with a dark band at the leading edge, whereas *C. cereale* is a hemibiotroph producing lesions that can start dark brown and become pale, with black dots (acervuli) that release the next generation of conidiospores [31]. The second leaf from each plant was harvested 11 days after pathogen inoculations and immediately placed in a sterile centrifuge tube on ice – after harvesting all experimental leaves, samples were moved to a −80°C freezer for storage until DNA extraction.

## DNA extraction and sequencing

DNA extraction was conducted on 94 randomly selected samples from all treatment groups. These samples were retrieved from a −80°C freezer and immediately placed in liquid nitrogen for one minute, followed by manual grinding with sterile pestles. Once ground into a fine powder, each leaf sample was returned to the −80°C freezer. DNA was extracted from these samples using the DNEasy PowerSoil kit, following the manufacturer's instructions (Qiagen). The extracted DNA samples were then stored at −80°C until library preparation.

For the preparation of fungal libraries, we used the Zymo Quick-16S NGS Library Prep Kit, customized with ITS1-F and ITS2 fungal primers [32,33]. Bacterial libraries were constructed using the same Zymo Quick-16S NGS Library Prep Kit, but targeting the V3-V4 regions of the 16S rRNA gene. This kit facilitated the amplification of bacterial DNA, attachment of index primers, and subsequent quantification and pooling of samples. To enhance the specificity of bacterial sequencing, we used PCR blockers (PNA Bio Inc.) which reduce the amplification of host mitochondrial and chloroplast DNA during 16S sequencing, thereby increasing the proportion of informative bacterial sequences in our dataset. Each sample well had a final concentration of 1µM mPNA and pPNA. All sequencing was performed by the High Throughput Sequencing Facility at UNC using the Illumina MiSeq platform. Additionally, phiX was added to both the bacterial (1% phiX) and fungal (20% phiX) libraries to enhance sample heterogeneity and improve sequencing quality.

## Fungal and bacterial community analysis

Raw fungal and bacterial sequence reads were demultiplexed using the Illumina bcl2fastq pipeline (v.2.20.0) and sequence adapters were removed in QIIME2 using Cutadapt (version 2.9). The assignment of fungal and bacterial ASVs to sequencing outputs was done using DADA2 [34], with sequences below a quality score of 15 being excluded. Fungal taxa were identified by matching sequences to the UNITE fungal ITS database (version 8.99, released on April 2nd, 2020) in QIIME2 [35], while bacterial taxa were matched to the SILVA 16S database [36]. To explore diversity patterns across hormone treatments, community metrics such as richness and Shannon diversity were calculated in QIIME2 for both 16S and ITS reads, filtering out all reads classifying as *R. solani* and *C. cereale*. Additionally, samples with fewer than 1,000 reads were excluded, resulting in 57 samples analyzed for bacterial Shannon diversity, richness, and Bray-Curtis dissimilarity, and 92 samples analyzed for fungal Shannon diversity, richness, and Bray-Curtis dissimilarity (S1 Table). The code used for this analysis is available here: https://github.com/juliegeyer/Microbiome/blob/main/QIIME2%20Code

## Filtering 16S host reads

Due to the non-specificity of 16S V3-V4 primers, some reads mapped to host mitochondrial or chloroplast DNA. Host reads were removed prior to analyzing bacterial communities using Bowtie2 (to align reads to tall fescue chloroplast and mitochondrial reference genomes), and Kraken2 (to remove host-associated reads). After this process, no sequencing reads mapped to the host. For both 16S and ITS reads, samples with fewer than 1,000 reads were filtered out. Then, all samples were rarefied using the QIIME2 'alpha-rarefaction' plugin to a uniform depth corresponding to the lowest read count in each dataset. Specifically, ITS data were rarefied to 6,000 reads per sample, while 16S data were rarefied to 1,000 reads per sample.

## Statistical analyses

**Effects on diversity of microbial taxa.** We modelled the effects of the hormone treatment and pathogen exposure and their interaction on microbial diversity and richness using a linear model with the independent variables in the form: *Hormone Treatment + Inoculation*. Separate models were run for fungi and bacteria to understand how each group responded to experimental treatments. We were unable to distinguish between the effects of inoculation and symptomatic infection in our models due to the low sample size of symptomatically infected host individuals (Table 1). To evaluate the experiment's statistical power to detect an effect of hormone treatment on bacterial richness, given the final sample size in each treatment, we ran a post-hoc power analysis using the 'pwr.f2.test' function in the pwr package 1.3 in R. In this function, we specified the numerator and denominator degrees of freedom from the model analyzed (1 and 52,

**Table 1. Symptomatic rates of *R. solani* and *C. cereale*.** Inoculation with *C. cereale* produced symptoms most often when co-inoculated with *R. solani*, and inoculation with *R. solani* led to symptomatic infection most often when co-inoculated with *C. cereale*. Bold numbers represent the number of plants in each treatment combination that were symptomatic, and numbers in parentheses represent the total number of plants in each treatment group.

| | | *R. solani* Symptomatic Rate | | *C. cereale* Symptomatic Rate | |
|---|---|---|---|---|---|
| | | **Inoculation Treatment** | | **Inoculation Treatment** | |
| | | *Single* | *Co* | *Single* | *Co* |
| Hormone Treatment | *Control* | **4** (18) | **7** (11) | **0** (18) | **5** (11) |
| | *Salicylic Acid* | **6** (19) | **6** (11) | **0** (19) | **1** (11) |

respectively), the cut-off for statistical significance (alpha = 0.05), and the desired power of the test (probability = 0.8 of detecting a true effect), then the function estimated the minimum effect size (f2, a projected value of $R^2$) that the experiment could detect given those constraints.

### Effects on community composition of microbial taxa

To assess the impact of hormone treatment and pathogen exposure on bacterial and fungal community structure, Bray-Curtis dissimilarity was calculated for both 16S and ITS reads using QIIME2. Patterns in community dissimilarity were ordinated and visualized as principal coordinate analysis (PCoA) [37]. Permutational multivariate analysis of variance (PERMANOVA) was used on Bray-Curtis distances to test if differences in community structure were related to the hormone or inoculation treatment – each of these independent variables (hormone treatment and inoculation treatment) were implemented in separate models for both bacteria and fungi. The PERMANOVA was run in R using the 'adonis2' function in the vegan package 2.6–4.

### Effects on abundance of microbial taxa

To analyze changes in the relative abundance of bacterial and fungal genera based on hormone treatment and inoculation type, taxonomy counts (computed using QIIME2) were incorporated into multivariate generalized linear models (using the 'manyglm' function) with a negative binomial distribution. Separate models were developed for bacteria and fungi, with inoculation treatment and hormone treatment each serving as independent variables in their respective analyses. Our models were limited to include the 20 most abundant fungal and bacterial genera, while also filtering out unidentified taxa, to increase our ability to detect changes in specific genera as per Mason [38]. Multivariate GLMs assessed genus-level (univariate) responses to differences in hormone treatment and inoculation treatment. Univariate tests were adjusted for multiple comparisons using a Holm step down procedure. GLMs were run using the mvabund package (version 4.2.1) in R [39]. Code for all diversity analyses can be found here: https://github.com/juliegeyer/Microbiome/blob/main/salicylic_acid_coinfection_16s_its

## Results

Among the samples analyzed for fungal diversity and community structure, Illumina sequencing produced 5,180,787 reads, with 3,420,596 passing quality filtering, resulting in an average of 36,389 reads per sample. DADA2 identified 600 unique amplicon sequence variants (ASVs), which were classified as fungal taxa. The most abundant fungal genera detected were *Cladosporium spp.*, *Fibulochlamys spp.*, and *Talaromyces spp.* For bacterial diversity and community structure analysis, Illumina generated 633,301 microbial reads (after excluding host DNA), with 380,139 reads passing quality filters. This yielded an average of 3,426 reads per sample before filtering out those with fewer than 1,000 reads. After this filtering step, the average increased to 5,271 reads per sample. A total of 960 unique bacterial ASVs were identified, with the most abundant being *Microbacterium spp.*, *Methylobacterium spp.*, and *Pseudomonas spp.*

Across hormone treatment groups, inoculations of *R. solani* produced symptoms in about 30% of inoculated plants, and *C. cereale* produced symptoms in roughly 10% of inoculated plants. *C. cereale* inoculation led to symptomatic infection most often when it was co-inoculated with *R. solani* and when plants were treated without salicylic acid. More specifically, *C. cereale* never produced symptomatic infection when inoculated alone (n = 37) and caused symptoms in only one of eleven plants when co-inoculated with *R. solani* in the salicylic acid treatment group. Inoculation with *R. solani* produced symptoms more often when it was co-inoculated with *C. cereale* (Table 1). This facilitative effect of *C. cereale* on *R. solani* even when *C. cereale* was asymptomatic is consistent with a previous study [40]. None of the mock-inoculated and mock co-inoculated plants (across hormone treatment groups) showed any visual symptoms of disease. Despite low symptomatic rates, all inoculated plants were included in their respective treatment groups for downstream analyses regardless of the presence of symptoms.

## Salicylic acid did not influence the diversity of the fungal or bacterial leaf microbiome

Although there was more variation in fungal diversity in plants that received a salicylic acid hormone treatment, there was no statistical difference in fungal Shannon diversity by hormone treatment group (MLR, $F_{1,87}=1.94$, p=0.17) (Fig 2A). Likewise, hormone treatment did not predict differences in bacterial diversity (MLR, $F_{1,52}=0.55$, p=0.46) (Fig 2B). In addition, we also found that hormone treatment did not influence fungal richness (MLR, $F_{1,87}=1.52$, p=0.22) or bacterial richness (MLR, $F_{1,52}=1.14$, p=0.29) (S1 Fig). This test for an effect of hormone treatment on bacterial richness was estimated to have an 80% probability of detecting as significant an effect with an $R^2$ value as low as 0.151 (post-hoc power analysis; f2=0.151), indicating that our experiment had adequate power to detect effects of even modest size.

## Salicylic acid did not influence the community composition of fungi or bacteria on plant leaves

Fungal communities in and on plants that were treated with salicylic acid were indistinguishable from plants that were treated with a control treatment (multivariate PERMANOVA, $F_{1,87}=0.52$, p=0.95, $R^2=0.006$) (Fig 3A); likewise, bacterial community composition was not significantly affected by hormone treatment (multivariate PERMANOVA, $F_{1,52}=0.75$, p=0.79, $R^2=0.013$) (Fig 3B).

## Salicylic acid application did not alter the relative abundance of fungal or bacterial taxa

The relative abundance of the 20 most abundant fungal or bacterial taxa did not differ by hormone treatment (Fig 4A). After adjusting for multiple comparisons in a univariate test (Holm step down procedure) neither fungal (univariate GLM (0/20), p>0.05) nor bacterial (univariate GLM (0/20), p>0.05) genera differed in community structure as a function of hormone treatment (Fig 4B).

## Inoculation treatment influenced fungal, but not bacterial diversity

When considering fungal diversity, there tended to be more variation in both the mock co-inoculated plants, as well as the co-inoculated plants. Inoculation treatment altered fungal Shannon diversity (MLR, $F_{1,87}=2.72$, p=0.0496). Specifically, there were differences in fungal diversity between the single inoculation group, and the co-inoculation group (Tukey's HSD, p=0.03) – fungi were more diverse in plants treated with co-inoculations than plants treated with single inoculations

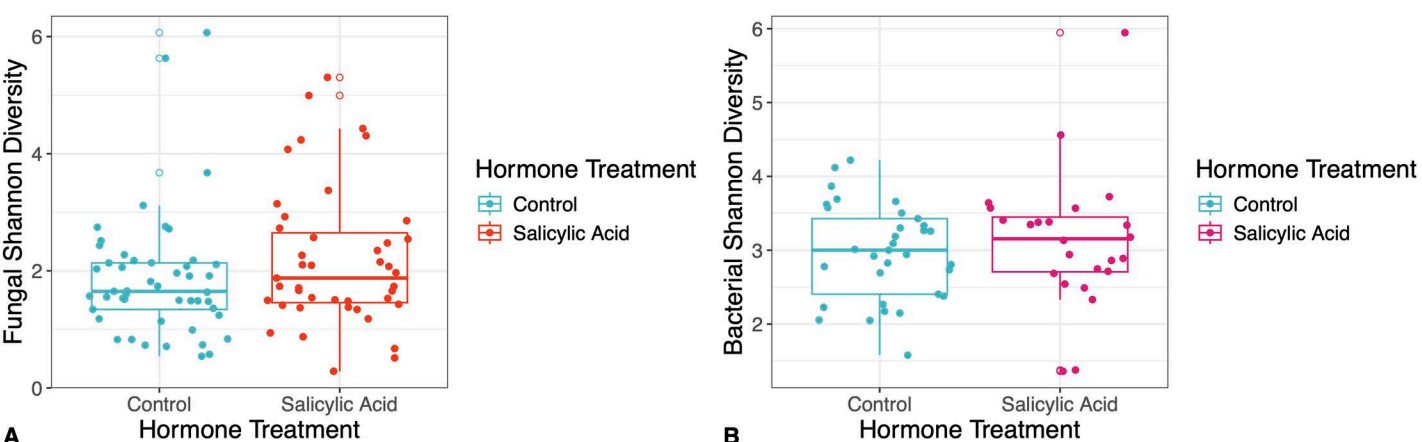

**Fig 2. Salicylic acid did not influence the diversity of the fungal or bacterial leaf microbiome across inoculation treatment groups.** Panels show (A) the application of salicylic acid did not influence the Shannon diversity of fungi on plant leaves (p=0.17) and (B) the application of salicylic acid did not influence the Shannon diversity of bacteria on plant leaves (p=0.46).

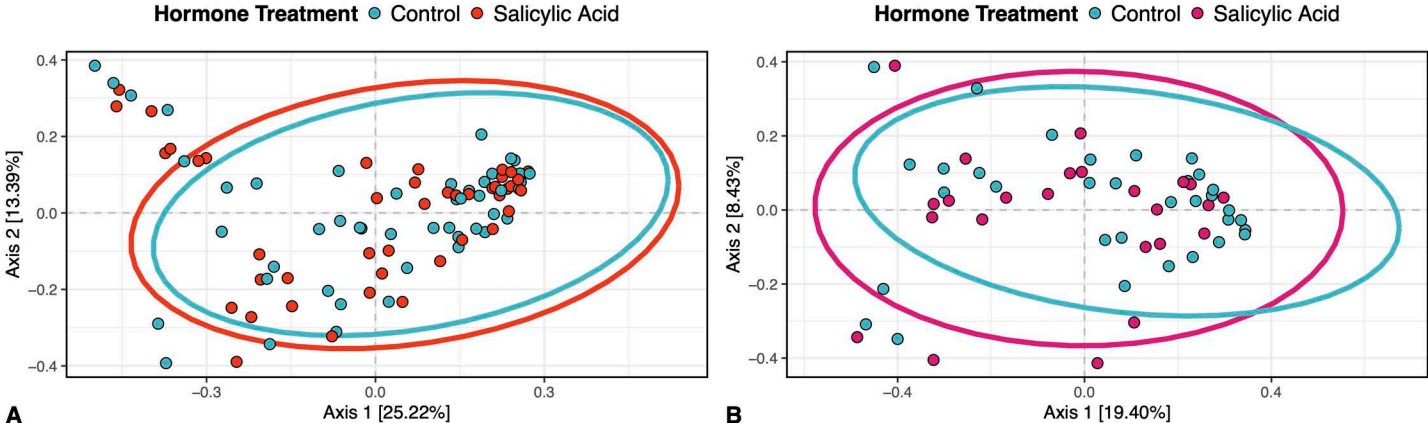

**Fig 3. Salicylic acid did not influence the community composition of fungi or bacteria on plant leaves across inoculation treatment groups.** Panels show (A) the application of salicylic acid did not influence the Bray-Curtis dissimilarity of fungi on plant leaves (p = 0.95) and (B) the application of salicylic acid did not influence the Bray-Curtis dissimilarity of bacteria on plant leaves (p = 0.79).

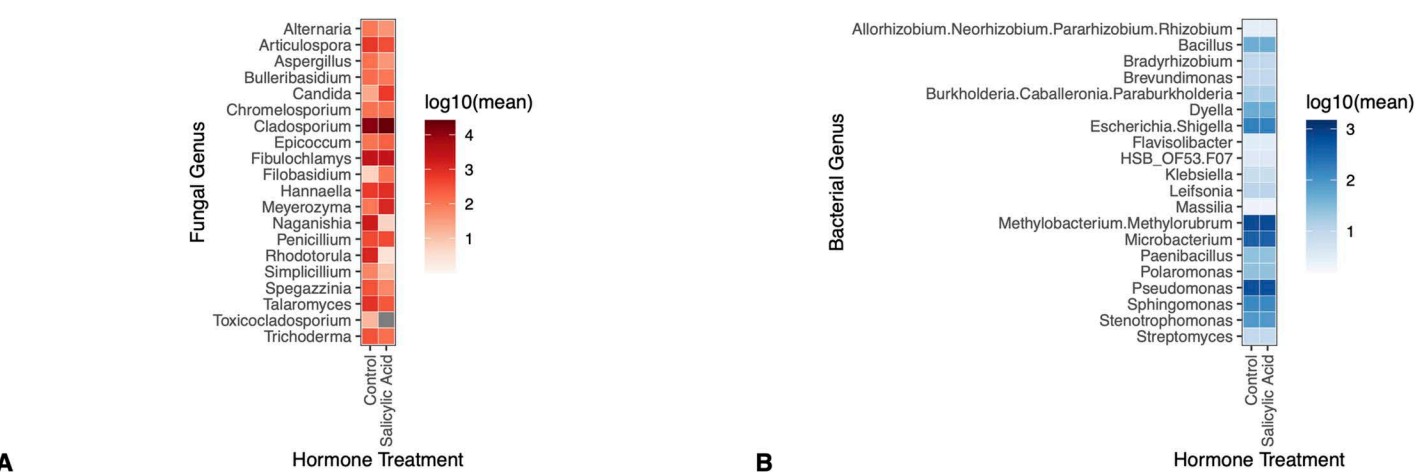

**Fig 4. Salicylic acid application did not alter the relative abundance of fungal or bacterial taxa across inoculation treatment groups.** Panels show that (a.) none of the 20 most abundant fungal genera were significantly different in their relative abundance by hormone treatment group (univariate GLM (0/20), p > 0.05 (adjusted)) and (b.) none of the 20 most abundant bacterial genera were significantly different in their relative abundance by hormone treatment group (univariate GLM (0/20), p > 0.05 (adjusted)). Genera, like *Allorhizobium spp.* were denoted as multiple species by our classifier because the classifier could not differentiate between the closely related species – as such, the genus name appears as a combination of the 4 separate, closely related genera. *HSB_OF53.F07 spp.* is a common soil bacterium in the phylum Chloroflexi [41].

(Fig 5A). However, bacterial Shannon diversity did not differ by inoculation group (MLR, $F_{1,52} = 0.16$, p = 0.92) (Fig 5B). Inoculation treatment did not affect fungal or bacterial richness (S2 Fig).

### Inoculation treatment influenced the community composition of fungi, but not bacteria on plant leaves

Plants that were co-inoculated with *R. solani* and *C. cereale* had greater variation in fungal community composition; likewise, fungal community composition differed in plants with different inoculation treatments (multivariate PERMANOVA, $F_{3,87} = 1.60$, p = 0.021, $R^2 = 0.052$) (Fig 6A). However, despite the significant relationship between

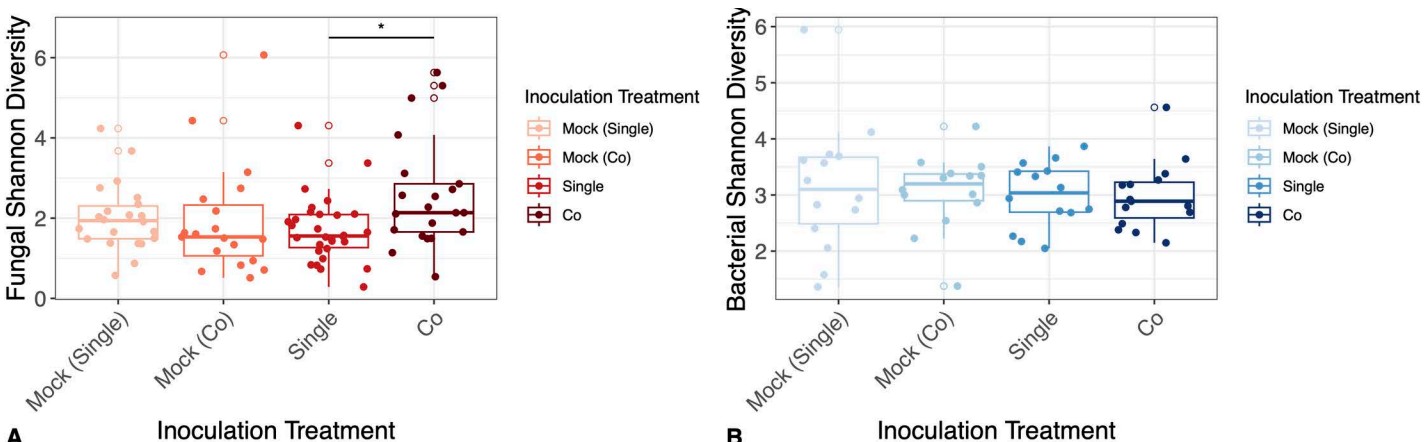

**Fig 5. Inoculation treatment influenced fungal, but not bacterial Shannon diversity across hormone treatment groups.** Panels show (A) inoculation treatment influenced fungal Shannon diversity (p = 0.0496) and (B) inoculation treatment did not influence bacterial richness (p = 0.92).

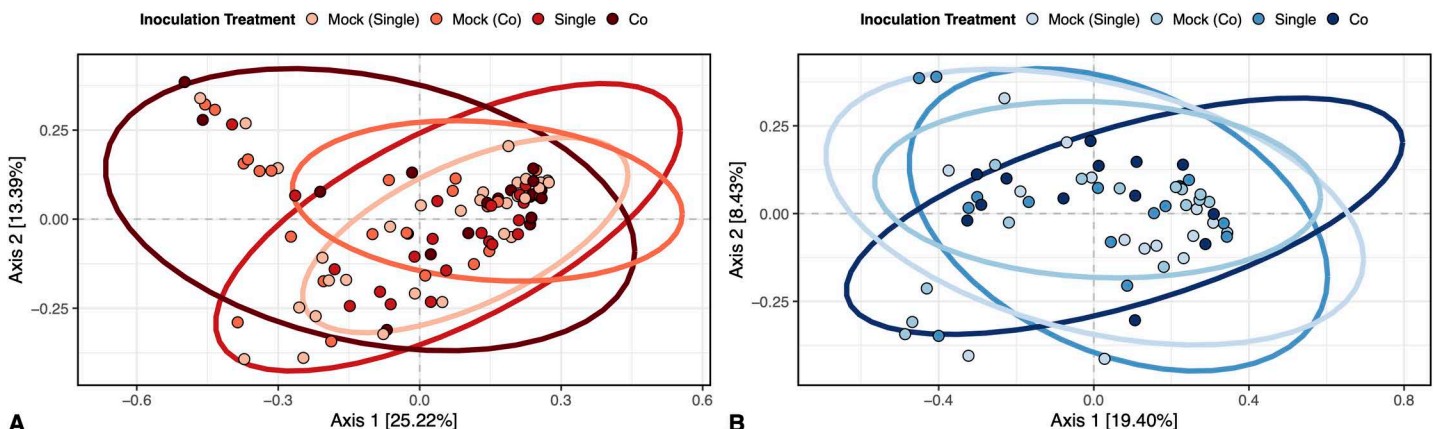

**Fig 6. Inoculation treatment influenced the community composition of fungi, but not bacteria on plant leaves across hormone treatment groups.** Panels show (A) inoculation treatment influenced the community composition of fungi on plant leaves (p = 0.021) and (B) inoculation treatment did not influence the community composition of bacteria on plant leaves (p = 0.47).

co-inoculation and fungal community composition, the amount of variation in fungal community composition explained by inoculation treatment remained low. The same trend was not observed for bacteria – inoculation treatment only had a weakly significant effect on the community composition of bacteria (multivariate PERMANOVA, $F_{3,52} = 0.100$, p = 0.47, $R^2 = 0.054$) (Fig 6B).

## Inoculation treatment did not alter the relative abundance of fungal or bacterial genera

The 20 most abundant fungal taxa, and the 20 most abundant bacterial taxa did not differ in abundance by inoculation treatment. After adjusting for multiple comparisons in univariate tests differences in fungal genera as a function of inoculation treatment were indetectable (univariate GLM, p > 0.05) (Fig 7A). Likewise, bacterial genera were not differentially abundant by inoculation treatment (univariate GLM, p > 0.05) (Fig 7B).

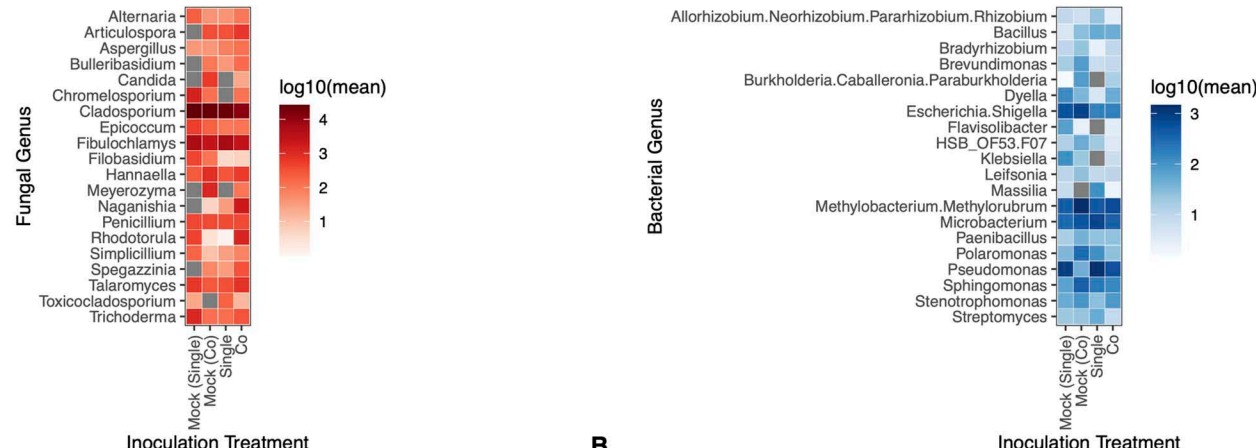

**Fig 7. Inoculation treatment did not alter the relative abundance of fungal or bacterial genera across hormone treatment groups.** Panels show that (A) none of the 20 most abundant fungal genera were significantly different in their relative abundance by inoculation treatment group (univariate GLM (0/20), p > 0.05 (adjusted)) and (B) none of the 20 most abundant bacterial genera were significantly different in their relative abundance by inoculation treatment group (univariate GLM (0/20), p > 0.05 (adjusted)). Genera, like *Allorhizobium spp.* were denoted as multiple species by our classifier because the classifier could not differentiate between the closely related species – as such, the genus name appears as a combination of the 4 separate, closely related genera. *HSB_OF53.F07 spp*. is a common soil bacterium in the phylum Chloroflexi [41].

## Discussion

Our study revealed that pathogen coinfection significantly reshaped fungal community composition and increased fungal diversity relative to single infections; however, the proportion of variation explained by coinfection was low. In contrast, salicylic acid application did not significantly alter the diversity, composition, or relative abundance of fungal or bacterial taxa. Bacterial communities remained unaffected by inoculation treatment.

Several mechanisms may explain why coinfection altered fungal communities while single infections did not. Coinfecting fungal pathogens may create localized tissue changes or alter the within-leaf environment in plant leaves despite no visible symptoms [42]. These changes may be more conspicuous if the co-inoculated pathogens act synergistically (instead of in isolation) to alter the biotic environment. This could favor increased diversity of decomposer fungi or select for fungi better adapted to the altered environment. Since fungi are the primary degraders of cellulose, which constitutes the majority of dry mass in grasses [43,44] this could explain why we observed changes (increases) in fungal diversity but not bacterial diversity.

The observed facilitative effect of *C. cereale* on *R. solani* in our study has been noted before within the tall fescue system [40] and provides a potential mechanism for microbial community shifts. The effect of co-inoculation on the diversity and community composition of fungi can produce a greater spatial extent of symptomatic infection (and therefore, potentially, a more altered biotic environment) in the co-inoculated group than the single inoculated group.

Unlike previous studies demonstrating a bi-directional relationship between hormone pathways and microbial communities [8,9,45], we discovered no effect of salicylic acid application on the fungal or bacterial microbiota with respect to community composition, diversity, or relative abundance of microbial taxa. To our knowledge, the effect of salicylic acid application on fungal communities in plant hosts has not yet been studied.

The discrepancy in results between our study and previous research on effects of salicylic acid could be attributed to how the hormone treatment was applied, or the specific host species examined. Previous studies have used the model organism, *Arabidopsis thaliana*, and salicylic acid production was manipulated by using knockout plants [15,16]. Although our system limited us to foliar salicylic acid application (tall fescue is non-model system without an annotated genome), foliar salicylic acid application has been shown in previous studies to modulate the expression of many genes, most of

which are involved in plant defense and growth [26,46]. Likewise, foliar application of salicylic acid has been shown to suppress the development of disease across pathogen feeding strategies, and across plant hosts [12,46,47] with salicylic acid concentrations as low as 0.1mM [27,28] and as high as 8mM [29]. Our intermediate application concentration of roughly 0.72mM had no effect on the bacterial or fungal communities on plant leaves suggesting that either the application method, timing, or host system may influence whether salicylic acid shapes the plant microbiome.

Our findings highlight several promising areas for future research. Our ability to disentangle the effects of symptomatic infection from inoculation treatment was constrained due to the limited number of symptomatic host plants. Future studies could investigate whether asymptomatic infection alters microbial communities in a manner similar to symptomatic infection – this may help resolve whether changes in the fungal communities are due to competition with resident fungi for resources, or whether these changes are the result of indirect ecological interactions or disruptions in the biotic environment.

The limited data on how host immune systems influence microbial communities highlights the need for further research, particularly regarding fungal communities. Additional research using different host systems and salicylic acid delivery methods could help resolve why our study found no effect of salicylic acid, in contrast to published studies. Specifically, future research could resolve this apparent discrepancy and expand our understanding of how plant hormones shape microbial communities by considering: 1) how foliar application versus systemic alteration of defense hormones alters microbial communities, and 2) capturing time-series data on how communities respond to hormone treatment, or infection over time.

Lastly, our study focused on community-level taxonomic trends, and therefore we sequenced only the ITS and 16S amplicons. To build on our study, future work could leverage metatranscriptomics and metagenomics to explore a mechanistic explanation for, and any functional implications of, the community shifts we observed.

## Supporting information

**S1 Table. The number of host individuals in each treatment group containing enough data (passed quality filtering) for analysis.** In total, 92 samples were analyzed for fungal diversity, community composition and abundance, and 57 samples were analyzed for metrics of bacterial community structure. Table abbreviations are as follows: "Mock (Single)" = plants that received a mock inoculation of *R. solani*; "Mock (Co)" = plants that received a mock co-inoculation of *R. solani* and *C. cereale*; "Single" = plants that received an inoculation of *R. solani*; "Co" = plants that received a co-inoculation of *R. solani* and *C. cereale*. Plants across inoculation treatment groups either received a hormone treatment of sterile water (control), or salicylic acid.
(TIF)

**S1 Fig. Salicylic acid did not influence the richness of the fungal or bacterial leaf microbiome across inoculation treatment groups.** Panels show (A) the application of salicylic acid did not influence the richness of fungi on plant leaves (p = 0.22) and (B) the application of salicylic acid did not influence the richness of bacteria on plant leaves (p = 0.29).
(TIF)

**S2 Fig. Inoculation treatment did not influence fungal or bacterial richness across inoculation treatment groups.** Panels show (A) inoculation treatment did not influence fungal richness (p = 0.14) or (B) inoculation treatment did not influence bacterial richness (p = 0.55).
(TIF)

## Acknowledgments

We thank Ezekiel Snyder and Bella Stiver for assistance with fungal inoculations, and Seth O'Conner for help with initial processing of data. Corbin Jones also provided invaluable guidance on library preparation and empirical depletion of host-associated DNA. We also wish to thank Jeff Roach who provided guidance on the bioinformatics workflow for

analyzing microbiome datasets. We appreciate the insightful commentary and feedback provided by Alecia Septer, Maggie Wagner, Corbin Jones, and Joel Kingsolver.

## Author contributions

**Conceptualization:** Julie K. Geyer, Rita L. Grunberg, Charles E. Mitchell.

**Data curation:** Julie K. Geyer.

**Formal analysis:** Julie K. Geyer, Rita L. Grunberg.

**Funding acquisition:** Charles E. Mitchell.

**Investigation:** Julie K. Geyer.

**Methodology:** Julie K. Geyer, Charles E. Mitchell.

**Resources:** Charles E. Mitchell.

**Visualization:** Julie K. Geyer.

**Writing – original draft:** Julie K. Geyer, Charles E. Mitchell.

**Writing – review & editing:** Julie K. Geyer, Rita L. Grunberg, Charles E. Mitchell.

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
