## [Decision Letter · Decision Letter 0]

4 Jun 2025

PONE-D-25-22997

Phyllosphere microbial communities are modulated by pathogen coinfection, but not a plant defense hormone

PLOS ONE

Dear Dr. Geyer,

Thank you for submitting your manuscript to PLOS ONE. After careful consideration, we have decided that your manuscript does not meet our criteria for publication and must therefore be rejected.

I am sorry that we cannot be more positive on this occasion, but hope that you appreciate the reasons for this decision.

Kind regards,

Eugenio Llorens

Academic Editor

PLOS ONE

Reviewers' comments:

Reviewer's Responses to Questions

**Comments to the Author**

1. Is the manuscript technically sound, and do the data support the conclusions?

Reviewer #1: No

Reviewer #2: Partly

2. Has the statistical analysis been performed appropriately and rigorously?

Reviewer #1: Yes

Reviewer #2: No

3. Have the authors made all data underlying the findings in their manuscript fully available?

Reviewer #1: Yes

Reviewer #2: No

4. Is the manuscript presented in an intelligible fashion and written in standard English?

Reviewer #1: Yes

Reviewer #2: No

Reviewer #1: The purpose of this study was to examine how sequential exposure to salicylic acid and a foliar fungal pathogen would influence the diversity, community structure, and differential abundance of both the fungal and bacterial communities of tall fescue. In general, the investigated treatments influenced neither the bacterial or fungal community structure nor differential abundance. The initial conclusion of the discussion is not well supported by the results. Fungal community composition variation was low in the co-inoculated (line 400-404); not “significant changes” as stated in the discussion. Whether fungi and bacteria interact with each other was not directly examined by this study and such should be removed from the discussion. The authors suggest that their fungal pathogens create necrotic lesions which would be favored by decomposer fungi and yet, most inoculated plants in this study remained asymptomatic. As for the findings from the salicylic acid treatment, the authors should have a better rationale for their chosen concentration. It’s possible that their concentration wasn’t biologically relevant to their host.

Reviewer #2: The manuscript suffers from several critical shortcomings that compromise its scientific rigor, clarity, and contribution to the field.

The following key issues highlight the major flaws in the study:

• The research question is poorly justified, and the study does not offer a novel contribution to the existing body of knowledge.

• The literature review is insufficient, lacking a comprehensive discussion of recent and relevant studies.

• The study appears to replicate previous findings without introducing significant innovation or new insights.

• The methodology is inadequately described, making replication difficult.

• The sample size is too small to draw meaningful conclusions, and statistical validation is weak.

• The experimental design lacks proper controls, leading to questionable reliability.

• Key parameters and variables are either not defined or inconsistently reported.

• The data analysis is superficial and does not provide a strong basis for the conclusions drawn.

• Statistical tests are either inappropriate or poorly applied, leading to misleading interpretations.

• The results are not presented logically, making it difficult to follow the study’s findings.

• The discussion section lacks depth and does not critically engage with the results in the context of existing literature.

• Figures and tables are not well-integrated into the discussion, and some data appear redundant or irrelevant.

• The authors overstate the significance of their findings without sufficient supporting evidence.

• The manuscript contains numerous grammatical and typographical errors, which hinder readability.

• The writing lacks coherence, and many sections are either repetitive or unclear.

• Formatting inconsistencies in citations, references, and section headings make the manuscript difficult to follow.

• The conclusion is vague and does not adequately summarize the key findings.

• The study's limitations are not acknowledged, giving a misleading impression of the research's reliability.

• No clear recommendations for future research are provided.

**Do you want your identity to be public for this peer review?** For information about this choice, including consent withdrawal, please see our Privacy Policy

Reviewer #1: No

Reviewer #2: No

- - - - -

---

## [Author Response · Author response to Decision Letter 1]

15 Jul 2025

Dear Dr. Llorens,

Thank you for reconsidering our manuscript following our appeal. We appreciate the journal's commitment to ensuring fair and thorough peer review.

As noted in our appeal, we identified substantial factual errors in Reviewer #2's assessment that indicated misunderstandings of our manuscript's content. We also raised concerns that this review appeared to be AI-generated based on its generic nature and inaccuracies. We identified one factual error in Reviewer #1's review regarding the interpretation of statistical significance, which may be why Reviewer #1 answered the journal’s review question #1 “No.” We are grateful that the journal agreed that these concerns warranted reconsideration.

Following the email guidance from the journal, we have prepared a revision that addresses Reviewer #1’s comments, and not those of Reviewer #2. We are confident that the revised manuscript, along with our detailed point-by-point response, demonstrates our commitment to producing rigorous scientific work that meets the journal's standards.

We look forward to your evaluation of our revised submission and appreciate the opportunity to improve our manuscript through this process.

Sincerely,

Julie Geyer

Response to Reviewer #1

Reviewer #1

The purpose of this study was to examine how sequential exposure to salicylic acid and a foliar fungal pathogen would influence the diversity, community structure, and differential abundance of both the fungal and bacterial communities of tall fescue. In general, the investigated treatments influenced neither the bacterial or fungal community structure nor differential abundance. The initial conclusion of the discussion is not well supported by the results. Fungal community composition variation was low in the co-inoculated (line 400-404); not “significant changes” as stated in the discussion.

Author: Thank you for taking the time to provide thoughtful and constructive feedback on our work. Your suggestions will help us improve the clarity of and strengthen our manuscript. We appreciate the reviewer’s concern to make sure that the manuscript’s conclusions are supported by the results. That said, we disagree with that “the initial conclusion of the discussion is not well supported by the results”. The finding that fungal community composition was affected by pathogen inoculation was supported by a standard and robust statistical test (PERMANOVA, F3,87 = 1.60, p = 0.021). However, the reviewer is correct that while we observed statistically significant differences in fungal community composition between treatments, the effect size was small (R² = 0.052). We have revised the discussion to highlight that while this treatment effect was statistically significant, it explained a small proportion of the variation in fungal community composition (lines 437-438 (track changes manuscript lines 448-449)); lines 440-443 (track changes manuscript lines 451-454)). We also updated the results section to explicitly note both the significant effects and the small effect size within the same sentence (lines 404-406 (track changes manuscript lines 415-417)).

Whether fungi and bacteria interact with each other was not directly examined by this study and such should be removed from the discussion.

Author: The reviewer makes a valid point. Our study design was not specifically structured to test direct fungal-bacterial interactions. We have revised the discussion to clarify that our observations suggest differential responses of fungal and bacterial communities to the same treatments, rather than speculating about fungal-bacterial interactions (deleted text starting at line 446 (track changes manuscript line 457).

The authors suggest that their fungal pathogens create necrotic lesions which would be favored by decomposer fungi and yet, most inoculated plants in this study remained asymptomatic.

Author: This is a valid point. The manuscript addressed it in the Discussion, suggesting effects of asymptomatic infection as a topic for future research (lines 481-486 (tracked changes manuscript lines 522-527)). That said, we acknowledge that our low symptomatic infection rates (approximately 30% for R. solani and 10% for C. cereale) limit our ability to connect necrotic lesion formation to the observed fungal community changes. We expanded the Discussion to further acknowledge this limitation and suggest that the observed effects may be due to either: (1) asymptomatic infections that still alter the within-leaf environment even without visible symptoms, or (2) localized tissue changes not visible at our assessment scale (lines 446-448 (tracked changes manuscript lines 457-459)).

As for the findings from the salicylic acid treatment, the authors should have a better rationale for their chosen concentration. It’s possible that their concentration wasn’t biologically relevant to their host.

Author: We agree that selecting biologically relevant treatments is an important part of the methods for any experiment, and we considered our selection of salicylic acid concentration thoroughly before conducting our experiment. Our manuscript addressed this topic in a full paragraph in the Discussion, but this reviewer comment indicates to us that we need to also address it in the Methods. The Discussion explained that we selected 0.72 mM salicylic acid based on previous literature showing effectiveness in this concentration range (0.1-8.0 mM) across diverse host species, and more specifically our concentration falls within the range that has demonstrated biological activity in other grass species (lines 472-475 (tracked changes manuscript lines 509-516)). The Discussion also acknowledged that the optimal concentration may be species-specific for our grass host (lines 465-467 (tracked changes manuscript lines 502-504)). Additionally, the Discussion suggested that future studies could examine differing host responses to foliar vs. systemic alteration of defense hormones (lines 490-493 (tracked changes manuscript lines 531-534)). To address the reviewer concern, we added text to our methods section to also include the underlying rationale for our decision to treat plans with 0.72 mM salicylic acid (lines 140-143; lines 147-150 (tracked changes manuscript lines 140-143; lines 147-150)).

---

## [Decision Letter · Decision Letter 1]

14 Oct 2025

Dear Dr. Geyer,

Thank you for submitting your manuscript to PLOS ONE. After careful consideration, we feel that it has merit but does not fully meet PLOS ONE’s publication criteria as it currently stands. Therefore, we invite you to submit a revised version of the manuscript that addresses the points raised during the review process.

We look forward to receiving your revised manuscript.

Kind regards,

Eugenio Llorens

Academic Editor

PLOS ONE

Journal Requirements:

2. Please include your tables as part of your main manuscript and remove the individual files. Please note that supplementary tables (should remain/ be uploaded) as separate "supporting information" files

Additional Editor Comments (if provided):

Reviewers' comments:

Reviewer's Responses to Questions

**Comments to the Author**

Reviewer #3: (No Response)

Reviewer #4: (No Response)

2. Is the manuscript technically sound, and do the data support the conclusions?

Reviewer #3: Yes

Reviewer #4: Yes

3. Has the statistical analysis been performed appropriately and rigorously?

Reviewer #3: Yes

Reviewer #4: Yes

4. Have the authors made all data underlying the findings in their manuscript fully available?

Reviewer #3: Yes

Reviewer #4: Yes

5. Is the manuscript presented in an intelligible fashion and written in standard English?

Reviewer #3: Yes

Reviewer #4: Yes

Reviewer #3: In this study Geyer, Grunberg and Mitchell, investigated how application of hormone salicylic acid and coinfection by two fungal pathogens influenced fungal and bacterial communities on leaves of Lolium arundinaceum. I believe that the experimental design is well developed and the results are in line with the methods and objectives of the study. However, I have some concerns, which are outlined below.

The authors do not clearly explain why they decided to use R. solani and C. cereale in their experiments. Is this co-infection possible in nature? Are they natural pathogens of L. arundinaceum? What other defenses besides chemical defense does this plant have to prevent infection by pathogens? Beyond the taxonomic identity of the fungal and bacterial species found, what information do the authors have about their biology and interactions in the context of a bacterial community?

I understand that this is an experimental manuscript, but I am concerned that both the approach of the manuscript and the discussion of the results do not take into account what occurs under natural conditions. Under natural conditions, how does infection occur in this plant species? Or in grasses in general? What are the implications in an ecological context for microorganism communities? Or in an evolutionary context for plant defense against herbivores?

It is not entirely clear why coinfection could modify microbial communities, compared to simple infections. At least, this is not explicitly stated. Including some predictions and hypotheses could help the reader.

Line 68, the idea is unclear.

Line 65, the authors attempt to explain the main idea of the manuscript, but it is confusing. Providing some examples of possible scenarios would give the reader more clarity.

Linea 85, It is difficult for me to understand why treatment with salicylic acid is the control treatment.

Methods

Line 95, Line 141, What does the control solution contain?

Line 265, Could you describe the characteristics of a symptomatic infection?

Reviewer #4: (No Response)

**Do you want your identity to be public for this peer review?** For information about this choice, including consent withdrawal, please see our Privacy Policy

Reviewer #3: No

Reviewer #4: No

---

## [Author Response · Author response to Decision Letter 2]

16 Dec 2025

Response to Reviewers

Dear Dr. Llorens,

Thank you for inviting us to revise our manuscript following our appeal. We thank the reviewers for their thoughtful and constructive feedback, which has substantially strengthened our manuscript.

In response to the comments from Reviewer 3, we have made several key improvements to the work. We have expanded the introduction to clarify the rationale for selecting R. solani and C. cereale as our focal pathogens, explaining that these species naturally co-occur and co-infect within the tall fescue system. We have also added text to further clarify the treatments in our factorial design, specifically the contents of the control hormone treatment. As suggested, we have added to our discussion by proposing several mechanisms for the shifts we see in fungal communities following coinfection. Lastly, we have acknowledged the limitations of our amplicon sequencing approach in the context of functional profiles. We believe these revisions have significantly improved the clarity and depth of the manuscript, and we appreciate the opportunity to address these important points.

We look forward to your evaluation of our revised submission and appreciate the opportunity to improve our manuscript through this process.

Reviewer #3

In this study Geyer, Grunberg and Mitchell, investigated how application of hormone salicylic acid and coinfection by two fungal pathogens influenced fungal and bacterial communities on leaves of Lolium arundinaceum. I believe that the experimental design is well developed and the results are in line with the methods and objectives of the study. However, I have some concerns, which are outlined below.

Author: Thank you for taking the time to provide thoughtful and constructive feedback on our work. We will address the concerns below.

The authors do not clearly explain why they decided to use R. solani and C. cereale in their experiments. Is this co-infection possible in nature? Are they natural pathogens of L. arundinaceum?

Author: This is a good point. We added information to the introduction (lines 81-83; lines 87-88) explaining that R. solani and C. cereale are naturally co-occurring and co-infecting pathogens within the tall fescue system.

What other defenses besides chemical defense does this plant have to prevent infection by pathogens?

Author: Grass species have several physical defenses hindering colonization by plant pathogens, including possessing a waxy cuticle. In our study, we decided to focus on inducible and dynamic defenses which have known interactions with microbial communities, rather than physical defenses. However, we have added information to lines 41-42 to acknowledge that immune hormones are just one component of the multifaceted plant immune system.

Beyond the taxonomic identity of the fungal and bacterial species found, what information do the authors have about their biology and interactions in the context of a bacterial community?

Author: This is an important question that is likely dependent on ecological/biological context. We are hesitant to speculate about the ecological function of these taxa as our study explicitly focused on community-level taxonomic trends, not functional profiles, and therefore we sequenced only the ITS and 16S amplicons. We have added additional text to lines 475-478 which acknowledges that future work could explore this question using metatranscriptomics or metagenomics.

I understand that this is an experimental manuscript, but I am concerned that both the approach of the manuscript and the discussion of the results do not take into account what occurs under natural conditions. Under natural conditions, how does infection occur in this plant species? Or in grasses in general? What are the implications in an ecological context for microorganism communities? Or in an evolutionary context for plant defense against herbivores?

Author: We agree that ecological context is important for interpreting our results. Thank you for pointing out that our manuscript needed more content about this context. In response, we added text on the pathogens’ natural modes of transmission (i.e., how does infection occur in this plant species) to manuscript lines 82-88 and text comparing these natural modes of transmission to our experimental inoculation methods on lines 186-188 and 193-195. As we selected our methods of experimental inoculation to be similar to each pathogen’s natural mode of transmission, we hesitate to speculate on the implications for microorganism communities or plant defense against herbivores.

It is not entirely clear why coinfection could modify microbial communities, compared to simple infections. At least, this is not explicitly stated. Including some predictions and hypotheses could help the reader.

Author: In our discussion, we hypothesize that an altered within-leaf environment due to coinfection could explain changes in fungal communities. We also discuss why this might explain changes in fungal communities, but not bacterial communities (lines 458-460). We have added text to lines 461-464 to explicitly propose potential mechanisms for the changes in diversity we observe in coinfections, but not single infections. Additionally, we have added text to the introduction to provide literature context supporting our hypothesis that coinfections differentially alter microbial communities compared to single infections (lines 74-78).

Line 68, the idea is unclear.

Author: The text on line 68 previously read:

“their microbes influence each other. Plants can modify their microbial communities”.

We appreciate the reviewer’s concern and changed the wording of this sentence to increase clarity (lines 65-67). The following sentences (lines 67-74) describe and cite studies which support this sentence.

Line 65, the authors attempt to explain the main idea of the manuscript, but it is confusing. Providing some examples of possible scenarios would give the reader more clarity.

Author: The text on line 65 previously read:

“The interaction between coinfecting pathogens and plant hormones has been”

We modified the wording of this sentence to increase clarity (line 67).

Linea 85, It is difficult for me to understand why treatment with salicylic acid is the control treatment.

Author: We re-worded this sentence to increase clarity (lines 96-97) – salicylic acid is our experimental treatment, and sterile water is our control treatment. We also added clarifying text to lines 107, 152, and 161 - 162.

Methods

Line 95, Line 141, What does the control solution contain?

Author: We have added additional text to lines 96-97, 107, and 152, and 161 - 162 to clarify earlier that our control solution is sterile water.

Line 265, Could you describe the characteristics of a symptomatic infection?

Author: we added text to lines 206-210 to describe the visual indicators of symptomatic infection.

---

## [Decision Letter · Decision Letter 2]

11 Jan 2026

Phyllosphere microbial communities are modulated by pathogen coinfection, but not a plant defense hormone

PONE-D-25-22997R2

Dear Dr. Geyer,

We’re pleased to inform you that your manuscript has been judged scientifically suitable for publication and will be formally accepted for publication once it meets all outstanding technical requirements.

Kind regards,

Eugenio Llorens

Academic Editor

PLOS One

Additional Editor Comments (optional):

Reviewers' comments:

Reviewer's Responses to Questions

**Comments to the Author**

Reviewer #3: All comments have been addressed

Reviewer #4: All comments have been addressed

2. Is the manuscript technically sound, and do the data support the conclusions?

Reviewer #3: Yes

Reviewer #4: Yes

3. Has the statistical analysis been performed appropriately and rigorously?

Reviewer #3: Yes

Reviewer #4: Yes

4. Have the authors made all data underlying the findings in their manuscript fully available?

Reviewer #3: Yes

Reviewer #4: Yes

5. Is the manuscript presented in an intelligible fashion and written in standard English?

Reviewer #3: Yes

Reviewer #4: Yes

Reviewer #3: I have no further comments on the manuscript. The authors address all my comments from the previous version.

Reviewer #4: (No Response)

**Do you want your identity to be public for this peer review?** For information about this choice, including consent withdrawal, please see our Privacy Policy

Reviewer #3: No

Reviewer #4: **Yes:** Dr. Muhammad Abrar

---

## [Editor Report · Acceptance letter]

PONE-D-25-22997R2

PLOS One

Dear Dr. Geyer,

I'm pleased to inform you that your manuscript has been deemed suitable for publication in PLOS One. Congratulations! Your manuscript is now being handed over to our production team.

Kind regards,

on behalf of

Dr. Eugenio Llorens

Academic Editor

PLOS One